# An evolutionary NS1 mutation enhances Zika virus evasion of host interferon induction

Hongjie Xia[1], Huanle Luo[2], Chao Shan[1], Antonio E. Muruato[1,2], Bruno T.D. Nunes[1,3], Daniele B.A. Medeiros[1,3], Jing Zou[1], Xuping Xie[1], Maria Isabel Giraldo[2], Pedro F.C. Vasconcelos[3,4], Scott C. Weaver[2,5,6,7,8], Tian Wang[2,7,9], Ricardo Rajsbaum[2,5] & Pei-Yong Shi[1,5,6,7,8,10]

Virus–host interactions determine an infection outcome. The Asian lineage of Zika virus (ZIKV), responsible for the recent epidemics, has fixed a mutation in the NS1 gene after 2012 that enhances mosquito infection. Here we report that the same mutation confers NS1 to inhibit interferon-β induction. This mutation enables NS1 binding to TBK1 and reduces TBK1 phosphorylation. Engineering the mutation into a pre-epidemic ZIKV strain debilitates the virus for interferon-β induction; reversing the mutation in an epidemic ZIKV strain invigorates the virus for interferon-β induction; these mutational effects are lost in IRF3-knockout cells. Additionally, ZIKV NS2A, NS2B, NS4A, NS4B, and NS5 can also suppress interferon-β production through targeting distinct components of the RIG-I pathway; however, for these proteins, no antagonistic difference is observed among various ZIKV strains. Our results support the mechanism that ZIKV has accumulated mutation(s) that increases the ability to evade immune response and potentiates infection and epidemics.

[1] Department of Biochemistry & Molecular Biology, University of Texas Medical Branch, Galveston, TX 77555, USA. [2] Department of Microbiology & Immunology, University of Texas Medical Branch, Galveston, TX 77555, USA. [3] Department of Arbovirology and Hemorrhagic Fevers, Evandro Chagas Institute, Ministry of Health, Ananindeua, Pará State, Brazil. [4] Department of Pathology, Pará State University, Belém, Brazil. [5] Institute for Human Infections & Immunity, University of Texas Medical Branch, Galveston, TX 77555, USA. [6] Institute for Translational Science, University of Texas Medical Branch, Galveston, TX 77555, USA. [7] Sealy Center for Vaccine Development, University of Texas Medical Branch, Galveston, TX 77555, USA. [8] Sealy Center for Structural Biology & Molecular Biophysics, University of Texas Medical Branch, Galveston, TX, USA. [9] Department of Pathology, University of Texas Medical Branch, Galveston, TX 77555, USA. [10] Department of Phamarcology & Toxicology, University of Texas Medical Branch, Galveston, TX 77555, USA. Correspondence and requests for materials should be addressed to P.-Y.S. (email: peshi@utmb.edu)

Zika virus (ZIKV) is a newly re-emerging arbovirus that belongs to the *flavivirus* genus of the Flaviviridae family. Besides ZIKV, many flaviviruses are significant human pathogens, including yellow fever (YFV), dengue (DENV), West Nile (WNV), Japanese encephalitis (JEV), and tick-borne encephalitic viruses (TBEV). ZIKV was initially isolated from a sentinel rhesus macaque in 1947 in the Zika Forest of Uganda[1]. Until 2007, ZIKV had silently circulated in many parts of Africa and Asia without causing detected severe diseases or large out-breaks, with fewer than 20 documented human infections[2]. However, during the recent epidemics, ZIKV infection has caused devastating severe diseases, including congenital malformations in the fetus of infected pregnant women (microcephaly and fetal demise) and Guillain-Barré syndrome in adults[3–5]. Phylogenetic analysis revealed that ZIKV evolved long ago into African and Asian lineages[6]. Strains from the Asian lineage are responsible for the recent large-scale epidemics on the Yap Island in 2007, in the French Polynesia and South Pacific in 2013, and in the Americas in 2015 and 2016[7,8]. The recent emergence of ZIKV could be driven by a number of potential mechanisms, including the acquisition of genetic changes that increase its ability to infect in humans and mosquitoes[9,10]. Indeed, a single amino acid sub-stitution in ZIKV NS1 protein was recently shown to enhance viral infectivity for *Aedes aegypti* mosquitoes, which may have facilitated transmission during the recent epidemics[11]. Another mutation was more recently reported to increase fetal micro-cephaly in a mouse model[12]. Whether other mutations of ZIKV that can modulate mammalian immune response and disease outcome remains to be determined.

Innate immune response is the first line of host defense against viral infection. Multiple host pattern recognition receptors, including Toll-like receptors and retinoic acid-inducible gene I (RIG-I)-like receptors, detect different pathogen-associated molecular patterns and trigger the antiviral responses by produc-ing type-I interferons (IFNs)[13]. After flavivirus infection, the innate immune response is primarily triggered through the recognition of viral RNA by the cytosolic RIG-I and melanoma differentiation associated gene 5 (MDA5). RIG-I acts at an early stage of immune responses to most flaviviruses, whereas MDA5 functions at a late stage of immune responses in a virus-dependent manner[14–17]. Once sensing the cytoplasmic viral RNA, RIG-I or MDA5 changes conformation to expose its caspase activation and recruitment domain (CARD). The exposed CARD of RIG-I or MDA5 interacts with the CARD domain of the mitochondrial antiviral adaptor protein (MAVS)[18]. Multiple signaling components are then recruited to MAVS, resulting in activation of the inhibitor of kappa-B kinase epsilon (IKKε) and TANK binding kinase 1 (TBK1), which phosphorylate the interferon regulatory factor 3 (IRF3). The phosphorylated IRF3 translocates to the nucleus and drives the transcription of type-I IFN genes[14,19,20]. The secreted type-I IFNs (IFN-α and IFN-β) bind to the IFN receptor (IFNAR) in an autocrine and paracrine manner[21,22], and signal through the Janus kinase (JAK)–signal transducer and activator of transcription protein (STAT) pathway to trigger the expression of hundreds of IFN-stimulated genes (ISGs) with antiviral functions[23,24].

Flaviviruses have positive, single-strand genomic RNAs of about 11,000 nucleotides that encode three structural proteins [capsid (C), precursor membrane (prM), and envelope (E)] and seven non-structural proteins (NS1, NS2A, NS2B, NS3, NS4A, NS4B, and NS5). The structural proteins, together with viral genomic RNA, form virions. The non-structural proteins parti-cipate in viral replication, assembly, and evasion of the host immune system[25]. Different flaviviruses have evolved various strategies to evade host immune responses[26–30]. For ZIKV, it was shown that (i) the virus antagonizes the type-I IFN response

during infection of human dendritic cells; (ii) viral NS1 and NS4B inhibit type-I IFN production at the step of TBK1 complex formation; and (iii) viral NS5 antagonizes type-I IFN signaling by degrading human STAT2 via the proteasome[31–34]. It remains to be determined whether other viral factors also contribute to the evasion of innate immune response and, more importantly, whether epidemic ZIKV strains from clinical isolates have evolved to dampen the host immune activation to increase its infection, transmission via enhanced viremia, and/or disease severity.

Here we show that ZIKV non-structural proteins inhibit RIG-I-induced IFN-β production through distinct mechanisms: NS2A, NS2B, and NS4B act at the step of TBK1 phosphorylation; NS4A at the step of IRF3 phosphorylation; and NS5 at a step down-stream of IRF3 phosphorylation through an NS5/IRF3 interac-tion. In addition, we have identified a mutation in ZIKV NS1 associated with arrival in the Americas that confers the ability to inhibit IFN-β induction. Mechanistically, this evolutionary NS1 mutation promotes the binding of NS1 to TBK1, resulting in reduced levels of TBK1 phosphorylation and IFN-β expression.

## Results

**Overall experimental approach.** The goals of this study are to identify (i) ZIKV non-structural proteins that can antagonize host type-I IFN production and (ii) mutations in the recent epidemic strains that modulate ZIKV evasion of type-I IFN induction. To achieve these goals, we first screened non-structural proteins from a pre-epidemic, Cambodian strain ZIKV (FSS13025 isolated from a patient in 2010) for their abilities to suppress IFN-β expression. Next, we compared the ability of viral proteins from three different strains (African Dakar-41525, Cambodian FSS13025, and Puerto Rico PRVABC-59) to identify genetic mutations that could affect the antagonism of type-I IFN induction.

**ZIKV non-structural proteins antagonize IFN-β promoter activation.** We used an IFN-β promoter-driven luciferase reporter plasmid (pIFN-β-luc) to screen for individual ZIKV proteins that can antagonize IFN-β production (Supplementary Fig. 1a). HEK-293T cells were co-transfected with reporter plas-mid pIFN-β-luc, control plasmid phRluc-TK (to normalize transfection efficiency), and vector expressing individual viral non-structural protein fused with a C-terminal HA-tag. A pre-epidemic, Cambodian ZIKV strain FSS13025 was used to clone individual viral proteins. An empty vector and EGFP expression plasmid were included as negative controls. At 24 h post-trans-fection, cells were stimulated with the dsRNA analog poly I:C for 16 h. The cells were quantified for luciferase activity to indicate IFN-β promoter activation (Supplementary Fig. 1a). Five out of the seven non-structural proteins, including NS2A, NS2B, NS4A, NS4B, and NS5, significantly suppressed the activation of IFN-β promoter, with NS2A exhibiting the strongest inhibition (about 75%); expression of NS2B/3 proteins did not significantly sup-press the activation of IFN-β promoter (Supplementary Fig. 1b). Western blot using anti-HA antibody showed comparable intra-cellular expression levels of viral proteins (Supplementary Fig. 1c). These results suggest that ZIKV NS2A, NS2B, NS4A, NS4B, and NS5 proteins can antagonize IFN-β induction.

**Inhibition of distinct components from the RIG-I pathway.** Flaviviruses are known to activate IFN-β production through the RIG-I pathway[15,17]. We screened candidate components of the RIG-I pathway that could potentially be targeted by viral proteins for inhibition (Fig. 1). HEK-293T cells were co-transfected with IFN-β reporter plasmid, viral protein expression plasmid, and plasmid expressing individual components from the RIG-I pathway. The constitutively active form of RIG-I [RIG-I

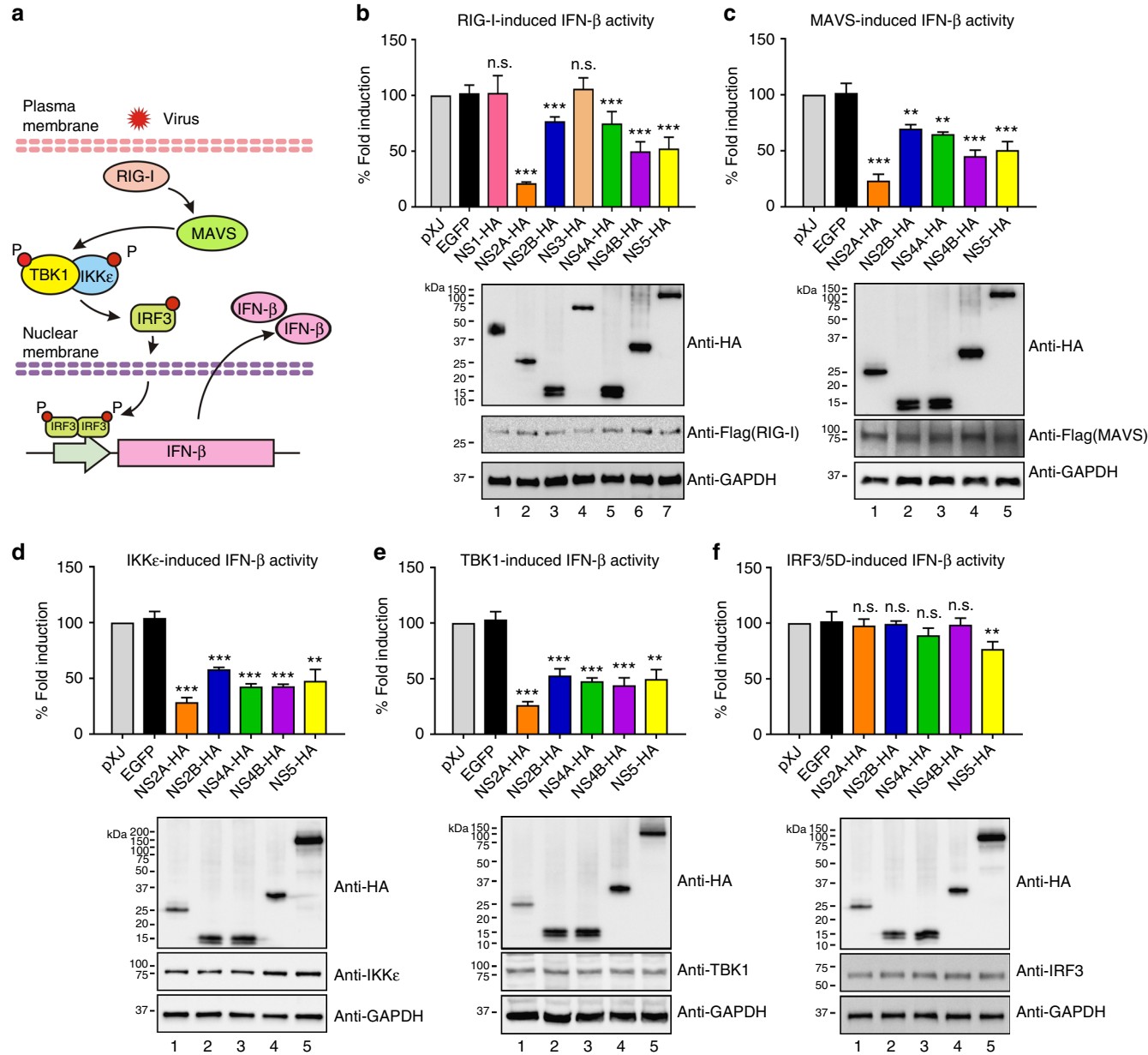

**Fig. 1** ZIKV non-structural proteins inhibit IFN-β production through the RIG-I pathway. **a** Schematics of the RIG-I mediated IFN-β production pathway. **b** IFN-β promoter luciferase activity assay. HEK-293T cells were co-transfected with IFN-β promoter firefly luciferase reporters, *renilla* luciferase plasmid, ZIKV non-structural protein-encoding plasmid or empty control plasmid, and stimulating plasmids RIG-I-(2CARD) (**b**), MAVS (**c**), IKKε (**d**), TBK1 (**e**), or IRF3/5D (**f**). All viral proteins were cloned from the pre-epidemic, Cambodian strain ZIKV (FSS13025) isolated from a patient in 2010. Data were normalized first by *renilla* luciferase values, and then normalized by none stimulated samples to obtain fold induction. Empty vector samples were set as 100% fold induction. Error bars represent mean ± SD. Results are representative of three independent experiments with each one in triplicate. Statistical significance was determined by Student's two-sided *t*-test, **$P < 0.01$, ***$P < 0.001$, or no significance (n.s.). Protein expression levels using western blot are shown below each panel. The original uncropped blots can be found in Supplementary Fig. 4

(2CARD), Flag-tagged], Flag-MAVS, Flag-IKKε, Flag-TBK1, or V5-IRF3/5D was individually transfected to activate the IFN-β promoter (Fig. 1a). At 24 h post-transfection, cells were assayed for luciferase activity. Similar to the results observed upon poly I: C treatment (Supplementary Fig. 1b), ZIKV NS2A, NS2B, NS4A, NS4B, or NS5 (but not NS1, NS3, EGFP, or empty plasmid vector) significantly suppressed IFN-β promoter activation when induced by ectopic expression of RIG-I (2CARD) (Fig. 1b), suggesting that these viral proteins antagonize IFN-β production through targeting cellular components at or downstream of RIG-I. The same five viral proteins also inhibited the activation of IFN-β promoter induced by MAVS, IKKε, or TBK1 (Fig. 1c–e).

However, only NS5, but not NS2A, NS2B, NS4A, and NS4B, weakly inhibited the activation (by about 23%) induced by IRF3/5D (a phosphomimetic form of IRF3; Fig. 1f). These results suggest that NS2A, NS2B, NS4A, and NS4B suppress IFN-β production through targeting TBK1 and/or IKKε kinases, whereas NS5 through targeting IRF3 or a downstream step (Fig. 1f).

**Inhibition of TBK1 and IRF3 activation.** Phosphorylation of TBK1, IKKε, and IRF3 is required for IFN-β mRNA induction[14]. We examined the effect of ZIKV NS2A, NS2B, NS4A, NS4B, and

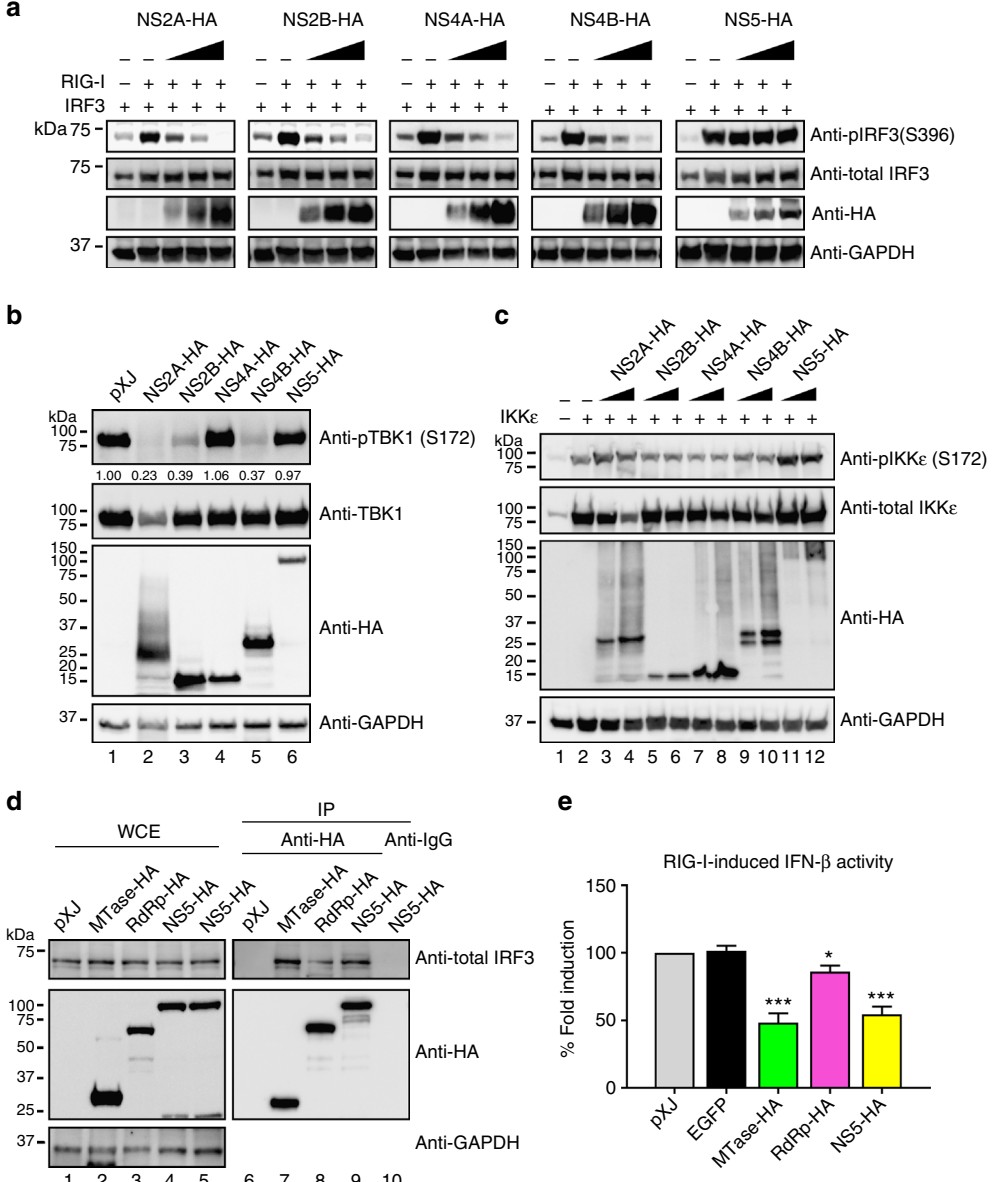

**Fig. 2** ZIKV non-structural proteins inhibit TBK1s and IRF3 activation. HEK-293T cells were co-transfected with IRF3-expressing plasmid together with RIG-I (2CARD)-coding plasmid (**a**), TBK1-expressing plasmid (**b**), or IKKε-expressing plasmid (**c**), as well as ZIKV non-structural protein-expressing plasmid or empty vector plasmid. Cells were harvested at 24 h post-transfection and subjected to western blotting with indicated antibodies, phosphorylated IRF3 (anti-pIRF3 S396), total IRF3 (anti-IRF3), phosphorylated TBK1 (anti-pTBK1 S172), total TBK1 (anti-TBK1), phosphorylated IKKε (anti-pIKKε S172), total IKKε (anti-IKKε), GAPDH (anti-GAPDH), and ZIKV proteins (anti-HA). All viral proteins were cloned from the pre-epidemic, Cambodian strain ZIKV (FSS13025) isolated from a patient in 2010. **d** Co-immunoprecipitation of NS5 and IRF3. HEK-293T cells were transfected with HA-fused MTase-, RdRp-, or full-length NS5-expressing plasmid and RIG-I(2CARD) stimulating plasmid. At 24 h post-transfection, cells were harvested and whole-cell extracts (WCE) were loaded as input (left panel). WCE were used for immunoprecipitation using anti-HA beads or anti-mouse IgG as a negative control, followed by detection of IRF3 using anti-total IRF3 antibody (right panel). The levels of proteins were quantitated by band intensity using Image lab software (BioRad). For **a**–**d**, the original uncropped blots can be found in Supplementary Fig. 5. **e** Luciferase assay of IFN-β promoter. MTase-, RdRp-, or NS5-expressing plasmid was co-transfected with RIG-I (2CARD) and reporter plasmids to HEK-293T cells. Cells were assayed for luciferase signals at 24 h post-transfection. Data are shown as mean ± SD from three independent experiments. Statistical significance was determined by Student's two-sided *t*-test, *$P < 0.05$, ***$P < 0.001$, or no significance (n.s.)

NS5 on the phosphorylation of TBK1, IKKε, and IRF3 (Fig. 2). HEK-293T cells were co-transfected with three plasmids: one expressing individual viral protein with an HA-tag; the second expressing IRF3; and the third expressing the constitutively activated form of RIG-I. At 24 h post-transfection, the phosphorylation status of TBK1, IKKε, and IRF3 was analyzed by western blot. The phosphorylation of IRF3 was inhibited by NS2A, NS2B, NS4A, or NS4B, but not NS5, in a dose-dependent

manner (Fig. 2a). The phosphorylation of TBK1 was suppressed by NS2A, NS2B, or NS4B, but not NS4A or NS5 (Fig. 2b). In contrast, none of these viral proteins affected IKKε phosphorylation (Fig. 2c). These results suggest that different ZIKV proteins antagonize IFN-β production through distinct cellular components from the RIG-I pathway: (i) NS2A, NS2B, and NS4B block TBK1 phosphorylation, (ii) NS4A suppresses IRF3 phosphorylation, and (iii) NS5 inhibits a step downstream of IFR3

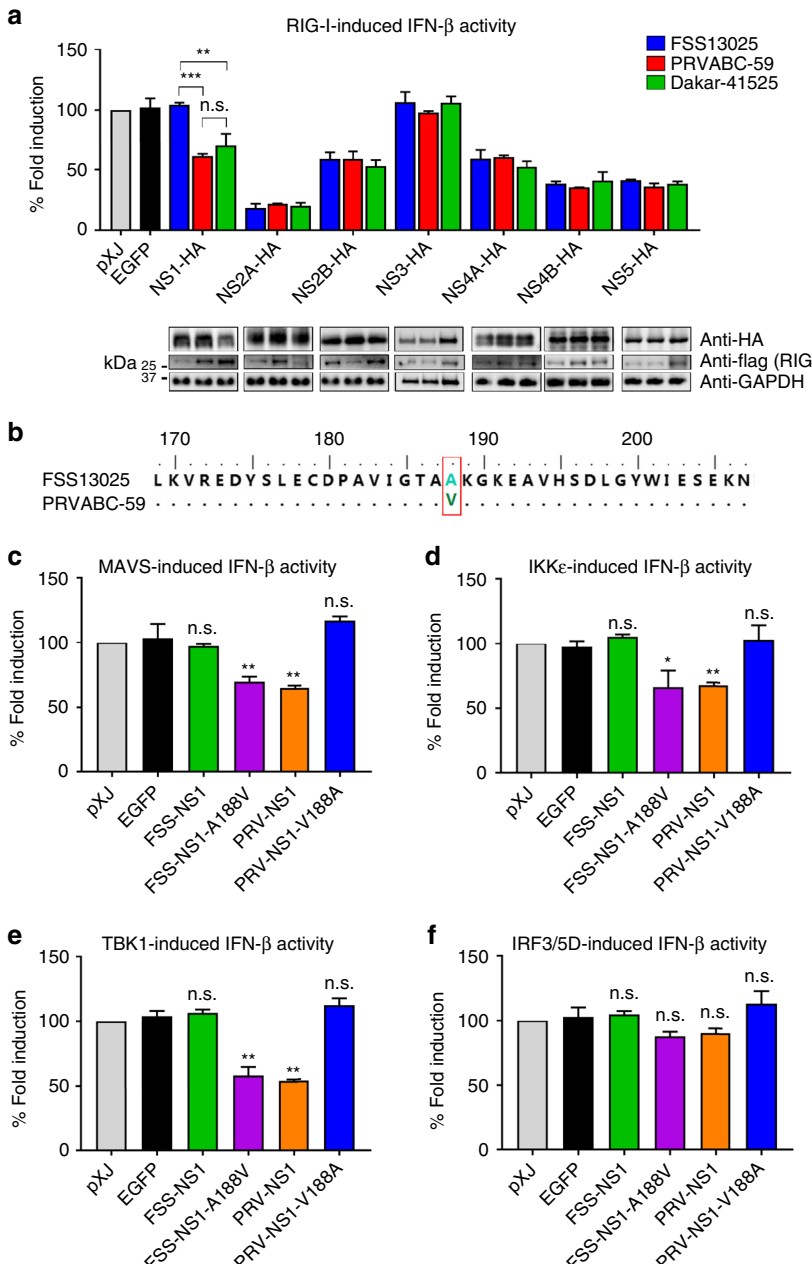

**Fig. 3** A single amino acid modulates NS1 to subvert IFN-β activation. **a** IFN-β promoter luciferase activity assay. HEK-293T cells were co-transfected with IFN-β promoter firefly luciferase reporter plasmid, *renilla* luciferase plasmid (for transfection normalization), non-structural proteins encoding plasmid from different ZIKV strains or control plasmid, and RIG-I (2CARD) expression plasmid as a stimulator. Cells were harvested and lysed at 24 h post-transfection, followed by luciferase assay as described in the legend to Fig. 1, and western blotting with indicated antibodies, the full blots can be found in Supplementary Fig. 6. **b** Amino acid sequence alignment between FSS13025 NS1 and PRVABC-59 NS1, residue 188 is highlighted with a red box. **c–f** HEK-293T cells were co-transfected with IFN-β promoter firefly luciferase reporter plasmid, *renilla* luciferase plasmid, ZIKV non-structural protein-encoding plasmid, or empty control plasmid, cells were stimulated by MAVS (**c**), IKKε (**d**), TBK1 (**e**), or IRF3/5D (**f**), followed by luciferase assay as described above. Results in (**a**, **c–f**) were normalized first by *renilla* luciferase values, and then normalized by none stimulated samples to obtain fold induction. Empty vector samples were set as 100% fold induction. Data are mean ± SD from three independent replicates, each experiment in triplicate. *P*-values were determined by unpaired Student's *t*-test, *$P < 0.05$, **$P < 0.01$, ***$P < 0.001$, or no significance (n.s.)

phosphorylation, possibly through IRF3 nuclear transportation or its binding to IFN-β promoter.

**Interaction of NS5 with IRF3.** To explore the molecular mechanism of NS5-mediated suppression of IFN-β induction, we examined if NS5 interacts with IRF3. Indeed, NS5 could pull down endogenous cellular IRF3 (Fig. 2d). Since flavivirus NS5 contains N-terminal methyltransferase (MTase) and C-terminal RNA-dependent polymerase (RdRp) domains, we tested the contributions of individual domains to the suppression of IFN-β induction. MTase (amino acids 1–264) and RdRp (amino acids 275–903) domain alone retained 82% and 51% of the full-length NS5-biding activity to IRF3, respectively (Fig. 2d).

Corroboratively, MTase domain inhibited IFN-β induction more efficiently than the RdRp domain (Fig. 2e). The results suggest that ZIKV NS5 suppresses IFN-β induction through binding to IRF3.

**A single mutation confers NS1 to antagonize IFN-β activation**. The above experiments were performed using proteins derived from a pre-epidemic ZIKV strain FSS13025 of the Asian lineage. We hypothesized that ZIKV strains with higher epidemic potential may have acquired mutations conferring stronger IFN antagonistic properties during infection. To test this hypothesis, we examined the abilities of viral non-structural proteins from two other ZIKV strains to antagonize IFN-β production in a RIG-I expression-induced activation assay: African lineage Dakar strain 41525 isolated in 1984 and Puerto Rico strain PRVABC-59 isolated during the ZIKV epidemics in 2015 (Fig. 3a). Remarkably, NS1 from both PRVABC-59 and Dakar-41525 strains suppressed IFN-β induction by 40% and 30%, respectively (Fig. 3a); in contrast, NS1 from the FSS13025 strain did not inhibit IFN-β activation (Supplementary Fig. 1b, Figs. 1b and 3a). No significant difference was observed for other viral proteins among the three ZIKV strains (Fig. 3a). Sequence analysis revealed only one amino acid difference in the NS1 protein between the FSS13025 and PRVABC-59 strains: an Ala has changed to a Val at position 188 (A188V; Fig. 3b). Compared with FSS13025, Dakar-41525 NS1 has a total of eight different amino acids, including the A188V change (Supplementary Fig. 2).

To examine the effect of NS1 A188V on antagonizing IFN-β induction, we tested four transiently expressed NS1 proteins in the IFN-β promoter luciferase assay: FSS13025 wild-type (WT) and A188V mutant, and PRVABC-59 WT and V188A mutant (Fig. 3c–f). FSS13025 A188V and PRVABC-59 WT NS1 proteins inhibited MAVS, IKKε, or TBK1-induced IFN-β promoter activation (Fig. 3c–e); the inhibitory effects were diminished when IFN-β activation was induced by IRF3/5D (Fig. 3f). In contrast, FSS13025 WT and PRVABC-59 V188A NS1 proteins were not able to antagonize IFN-β activation induced by MAVS, IKKε, TBK1, or IRF3/5D (Fig. 3c–f). The results suggest that 188-Val of NS1 is essential to subvert IFN-β activation, possibly through targeting the IKKε/TBK1 complex.

**NS1 with 188-Val antagonizes IFN-β production through TBK1**. To further define the molecular mechanism, we compared the effects of NS1 188-Ala or 188-Val protein on the phosphorylation of IRF3, TBK1, and IKKε (Fig. 4a–d). FSS13025 A188V NS1 and PRVABC-59 WT NS1 suppressed the phosphorylation of IRF3 in a dose-responsive manner (Fig. 4a, lanes 6–8 and Fig. 4b, lanes 3–5); these two proteins also inhibited the phosphorylation of TBK1 (Fig. 4c, lanes 3 and 4), but not the phosphorylation of IKKε (Fig. 4d, lanes 3 and 4). In contrast, FSS13025 WT NS1 and PRVABC-59 V188A NS1 did not affect the phosphorylation of IRF3, TBK1, or IKKε (Fig. 4a–d). These results indicate that NS1 with 188-Val inhibits IFN-β production through reducing TBK1 phosphorylation.

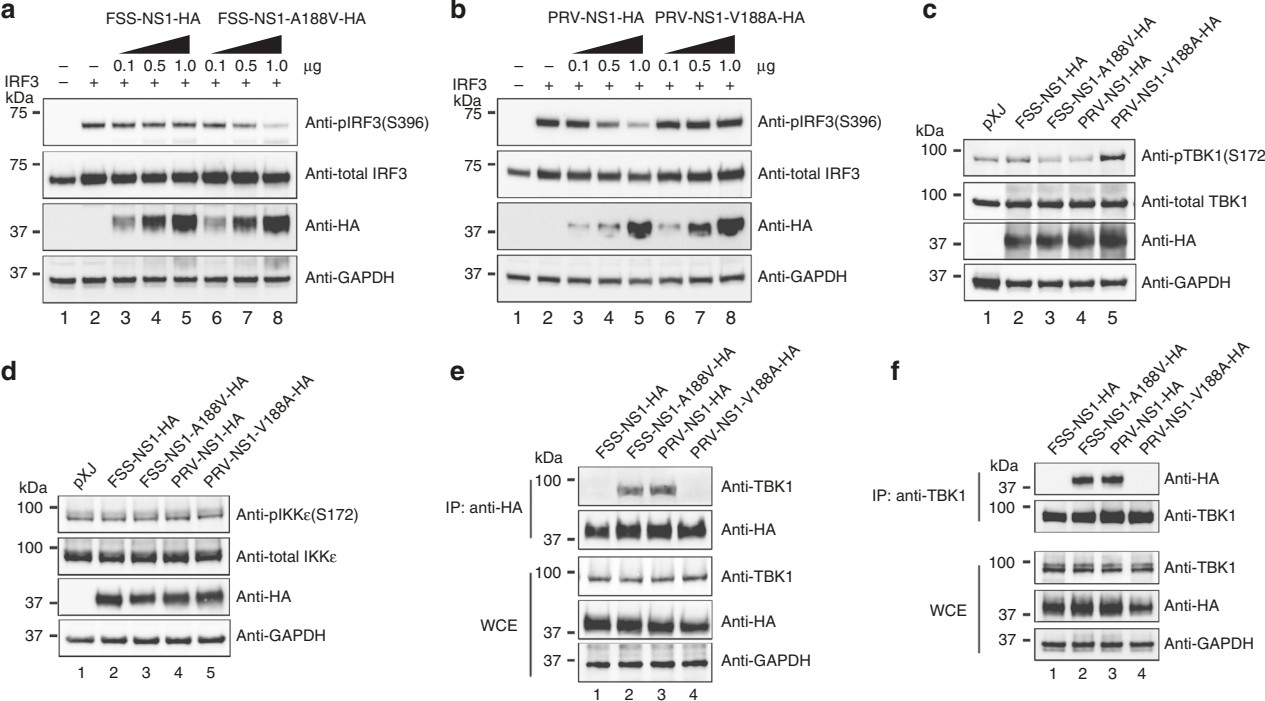

**Fig. 4** NS1 with 188-Val antagonizes IFN-β production through targeting TBK1. **a**, **b** HEK-293T cells were co-transfected with IRF3-expressing plasmid, RIG-I (2CARD)-encoding plasmid as a stimulator, FSS WT NS1-expressing plasmid or FSS NS1 A188V-expressing plasmid (**a**), PRV WT NS1-expressing plasmid or PRV NS1 V188A-expressing plasmid (**b**). At 24 h post-transfection, cells were harvested and lysed, and cell extracts were subjected to western blotting with indicated antibodies. **c**, **d** Plasmids expressing TBK1 (**c**) or IKKε (**d**), together with FSS13025 WT, FSS13025 A188V, PRVABC-59 WT, or PRVABC-59 V188A expression plasmids were co-transfected into HEK-293T cells. Cells were harvested at 24 h post-transfection and analyzed by western blotting for pTBK1 (anti-pTBK1 S172), total TBK1 (anti-TBK1), pIKKε (anti-pIKKε S172), total IKKε (anti-IKKε), GAPDH (anti-GAPDH), and NS1 protein (anti-HA). **e**, **f** Co-immunoprecipitation for detection of NS1 and TBK1 interaction. HEK-293T cells were transfected with TBK1-encoding plasmid and HA-tagged FSS NS1, FSS NS1 A188V, PRV NS1, or PRV NS1 V188A expression plasmid. At 24 h post-transfection, cells were harvested and whole-cell extracts (WCE) were used for immunoprecipitation using anti-HA beads (**e**) or anti-TBK1 antibody (**f**), followed by immunoblot with indicated antibodies. The original uncropped blots can be found in Supplementary Fig. 7

Next, we examined whether NS1 interacts with TBK1 and, if so, whether 188-Ala or 188-Val of NS1 changes the NS1/TBK1 interaction. Co-immunoprecipitation experiments showed that TBK1 was efficiently pulled down by NS1 with 188-Val (FSS13025 A188V and PRVABC-59 WT; Fig. 4e, lanes 2 and 3), whereas no TBK1 was pulled down by NS1 with 188-Ala (FSS13025 WT and PRVABC-59 V188A; Fig. 4e, lanes 1 and 4). Conversely, NS1 with 188-Val was efficiently co-immunoprecipitated by TBK1 (Fig. 4f, lanes 2 and 3), whereas

no NS1 with 188-Ala was pulled down by TBK1 (Fig. 4f, lanes 1 and 4). Collectively, the results indicate that NS1 with 188-Val binds to TBK1, such 188-Val NS1/TBK1 interaction reduces the phosphorylation of TBK1, leading to a decreased level of IFN-β production.

**Residue 188 of NS1 modulates IFN-β production in human cells.** We tested the biological relevance of the NS1 188-Ala and 188-Val in the context of ZIKV infection. Using the infectious

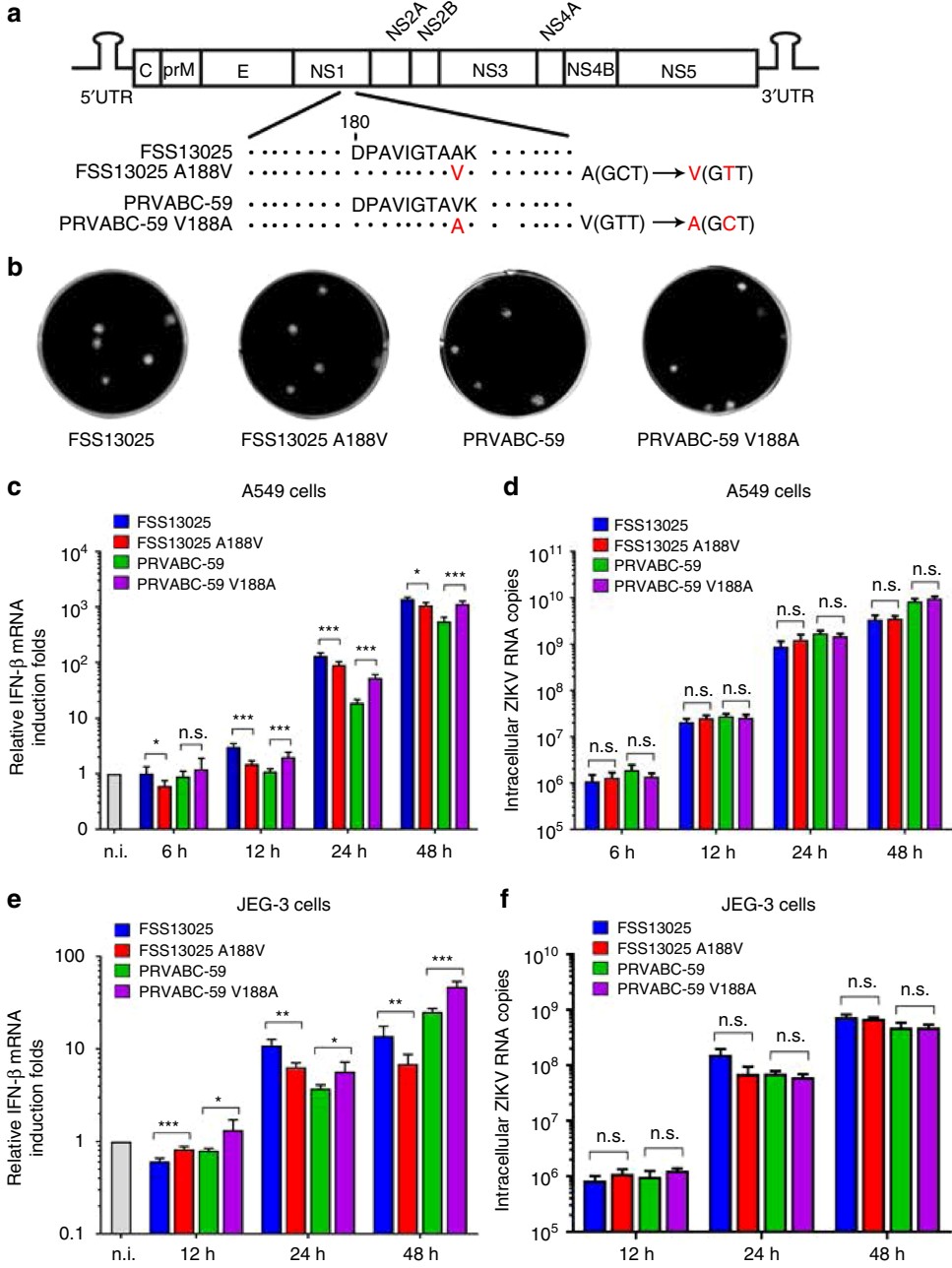

**Fig. 5** The identity of amino acid 188 of NS1 modulates IFN-β production during ZIKV infection of human cell lines. **a** Schematics of construction of ZIKV WT and corresponding NS1 mutant viruses. Codons for 188-Ala and 188-Val of NS1 are indicated. **b** Plaque morphology of recombinant viruses: FSS13025 WT, FSS13025 A188V mutant, PRVABC-59 WT, and PRVABC-59 V188A mutant. **c–f** IFN-β production of A549 cells (**c**) or JEG-3 cells (**e**) in response to ZIKV infection. A549 cells were infected with FSS13025 WT, FSS13025 A188V, PRVABC-59 WT, or PRVABC-59 V188A at an MOI of 0.5. Cells were harvested from 6 to 48 h p.i., total intracellular RNA was isolated, and *IFN-β* mRNA expression levels were quantified by qRT-PCR. *GAPDH* was used as a housekeeping gene for normalization. The mRNA levels are shown as fold induction over mock samples. **d** Intracellular viral RNA copies in A549 cells. **f** Intracellular viral RNA copies in JEG-3 cells. Error bars indicate standard deviations from three independent experiments. Statistical values were analyzed by unpaired Student's *t*-test, *$P < 0.05$, **$P < 0.01$, ***$P < 0.001$, or no significance (n.s.)

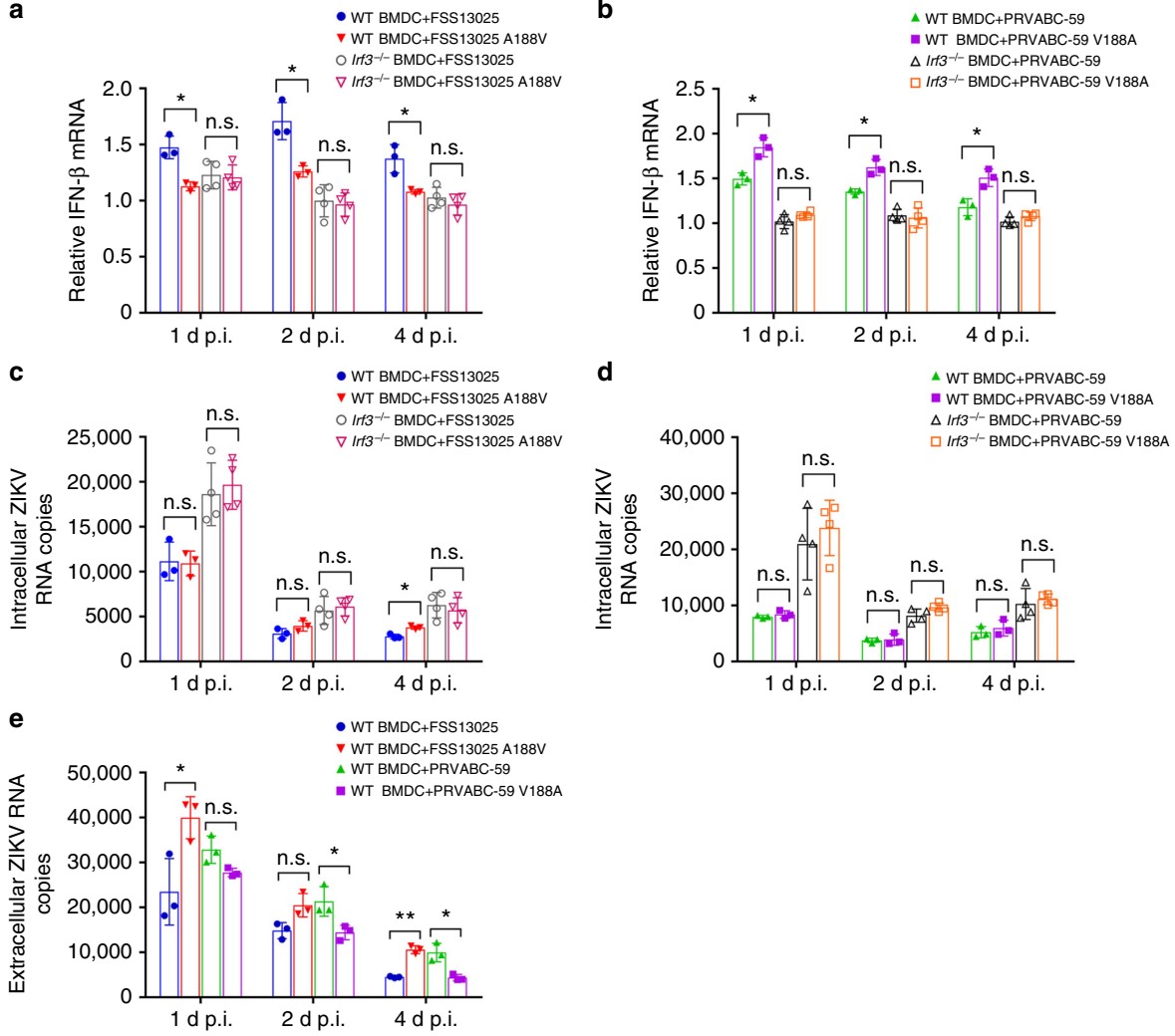

**Fig. 6** The role of NS1 mutation in evasion of IFN-β production in primary dendritic cells. BMDCs from C57BL/6J mice or $Irf3^{-/-}$ mice were infected with FSS13025 WT, FSS13025 A188V (**a**), PRVABC-59 WT, or PRVABC-59/V188A (**b**) at an MOI of 0.2. Cells were harvested and total intracellular RNA were isolated at days 1, 2, and 4 p.i., followed by qRT-PCR quantification of *IFN-β* mRNA (**a**, **b**) and intracellular viral RNA (**c**, **d**) as described in Fig. 5. **e** Extracellular viral RNA from C57BL/6J BMDC cells in (**a** and **b**) were measured by qRT-PCR. Data are presented as means ± SD from two independent experiments (with at least triplicates for each experiment). Statistical significance is presented as $*P < 0.05$, $**P < 0.01$, or no significance (n.s.) using unpaired Student's *t*-test

cDNA clones of FSS13025 and PRVABC-59[35,36], we prepared two pairs of recombinant viruses: FSS13025 WT and NS1 A188V mutant, and PRVABC-59 WT and NS1 V188A mutant (Fig. 5a). All four viruses exhibited similar plaque morphologies (Fig. 5b). On Vero cells that lack type-I IFN production, the A188V mutation did not affect viral replication (Supplementary Fig. 3a) or replicon RNA synthesis of strain FSS13025 (Supplementary Fig. 3b), suggesting that the NS1 mutation does not affect viral replication per se. Sequencing of the FSS13025 A188V and PRVABC-59 V188A viruses confirmed the engineered mutations without any other undesired changes.

Next, we compared the replication kinetics and the induction levels of IFN-β mRNA in two immune-competent human cell lines: A549 (human alveolar basal epithelial cells) and JEG-3 (human placenta trophoblastic cells). For the FSS13025 pair, the WT virus induced higher levels of IFN-β mRNA than the NS1 A188V virus in both cell lines (Fig. 5c, e), whereas no significant difference was observed for the intracellular viral RNA levels between the WT and mutant viruses (Fig. 5d, f). In contrast, for the PRVABC-59 pair, the WT virus induced lower levels of IFN-β

mRNA than the NS1 V188A mutant in both cell lines (Fig. 5c, e), whereas no significant difference was observed for the intracellular viral RNA levels between the WT and mutant viruses (Fig. 5d, f). The results indicate that the identity of amino acid 188 of NS1 modulates IFN-β production during ZIKV infection in human cell lines.

**The NS1 mutation affects IFN-β production in dendritic cells.** We infected bone marrow-derived dendritic cells (BMDCs) from WT C57BL/6J mice or from IRF3-knockout ($Irf3^{-/-}$) mice with WT and NS1 mutant viruses. In the C57BL/6J BMDCs, despite low levels of IFN-β mRNA expression (due to low infection rate of the immune-competent primary mouse cells), FSS13025 WT and PRVABC-59 V188A induced higher levels of intracellular IFN-β mRNA than their corresponding A188V mutant and WT viruses, respectively (Fig. 6a, b). In contrast, similar levels of intracellular IFN-β mRNA were detected in the $Irf3^{-/-}$ BMDCs infected with WT or mutant viruses of the FSS13025 or PRVABC-59 strain (Fig. 6a, b). Intracellular viral RNA levels were comparable between the WT and mutant viruses for each strain; as expected,

*Irf3*$^{-/-}$ BMDCs supported higher levels of viral RNA replication than WT C57BL/6J BMDCs did (Fig. 6c, d). Notably, the extracellular levels of viral RNA of FSS13025 A188V and PRVABC-59 WT were higher than those of corresponding WT and V188A, respectively (Fig. 6e). Collectively, these ex vivo experiments demonstrated the critical role of 188-Val in enabling NS1 to subvert IFN-β expression at a step upstream of IRF3 activation.

**IFN-β protein levels in infected mouse sera**. The A129 mice, defective in type-I IFN receptor, were commonly used to study ZIKV pathogenesis, therapeutics, and vaccine[37–39]. We infected 3-week-old A129 mice with equal amounts of WT or NS1 mutant ZIKV ($1 \times 10^5$ PFU) via the intraperitoneal route. Mock infection with phosphate-buffered saline was used as a control. Mouse sera were quantified for IFN-β protein expression and infectious virus using enzyme-linked immunosorbent assay (ELISA) and plaque assay, respectively. Increasing levels of IFN-β protein (Fig. 7a) and infectious virus (Fig. 7b) were detected from days 1 to 3 post-infection (p.i.). FSS13025 WT induced significantly higher levels of IFN-β protein than the corresponding NS1 A188V virus, whereas PRVABC-59 WT elicited lower levels of IFN-β protein than the corresponding NS1 V188A virus (Fig. 7a). Similar levels of viremia were detected for the WT and NS1 mutant viruses for FSS13025 or PRVABC-59 (Fig. 7b); this is not surprising because the A129 mice lack type-I IFN receptors to signal the antiviral effect. Collectively, the results demonstrate that infection with ZIKV containing the NS1 188-Val elicits a higher level of IFN-β protein than that with the NS1 188-Ala in the A129 mice.

**Neurovirulence in neonate mice**. It is important to compare the neurovirulence between the pre-epidemic FSS13025 and epidemic PRVABC-59 strains, and to determine whether the NS1 A188V mutation enhances the neurovirulence. So, we compared the neurovirulence of WT and NS1 A188V FSS13025 viruses as well as WT and NS1 V188A PRVABC-59. After intracranial injection of one-day-old CD-1 mice with $10^3$ PFU of virus ($n = 8$ per group), both PRVABC-59 WT and V188A viruses caused 100% deaths with average survival time (AST) of 13 and 12.9 days, respectively (Fig. 7c). In contrast, FSS13025 WT and A188V mutant caused 37.5% (3/8) and 62.5% (5/8) deaths with AST of 16 and 14.2 days, respectively (Fig. 7c); compared with FSS13025 WT, the A188V mutation significantly increased viral loads in the brains of infected mice on days 5–9 p.i. (Fig. 7d). The results suggest that (i) the epidemic PRVABC-59 is significantly more neurovirulent than the pre-epidemic FSS13025, (ii) the NS1 A188V mutation alone significantly increases the viral replication of FSS13025 strain in the brain of neonate CD-1 mice, and (iii) the NS1 V188A mutation alone does not affect the neurovirulence of PRVABC-59 strain. The different effects of NS1 188-Ala and 188-Val on neurovirulence in (ii) and (iii) are likely due to the differences in other viral proteins between the FSS13025 and PRVABC-59 strains, which may mask the phenotype of NS1 V188A in the context of PRVABC-59.

**Viremia and spleen viral loads in C57BL/6J mice**. Immunocompetent mice are required to examine the full effect of the NS1 mutation on viral replication through IFN production and signaling. Therefore, we infected 6-week-old C57BL/6J mice with $10^5$ PFU of WT FSS13025 and PRVABC-59 strains as well as their corresponding NS1 mutant viruses through the intraperitoneal route. Transient viremia and spleen viral loads were measured using qRT-PCR at days 1–3 post-infection. Consistent with the results from WT C57BL/6J BMDC (Fig. 6e), FSS13025 A188V and PRVABC-59 WT produced significantly higher viremia and spleen viral loads than the corresponding WT and

V188A mutant viruses, respectively (Fig. 7d, f). The results indicate that NS1 188-Val mutation enhances viral replication through suppressing IFN-β induction in the immunocompetent C57BL/6J mouse.

## Discussion

Understanding the mechanisms of epidemic ZIKV emergence, and its associated diseases, is critical to predict future risks as well as to target surveillance and control measures in key geographic locations[40]. In this study, we tested the hypothesis that the current epidemic strains of ZIKV have accumulated mutation(s) that enhances the viral antagonism of innate immune response, leading to increased viral replication and pathogenesis, and possibly more efficient mosquito transmission if viremia is enhanced in humans[35]. Compared with the pre-epidemic strains from Asian lineage, the epidemic strains isolated after 2012 have undergone an A188V mutation in the NS1 gene[11] that enabled the protein to suppress IFN-β induction. This mutational effect was consistently observed in cell culture, ex vivo BMDCs, A129 mouse, and immunocompetent C57BL/6J mouse (Figs. 5–7). Mechanistically, the A188V mutation renders the binding of NS1 to TBK1; such NS1/TBK1 interaction decreased TBK1 phosphorylation, resulting in reduced IRF3 phosphorylation and consequently IFN-β induction (Figs. 3 and 4). This mechanism was further supported by the loss of enhancement function of 188-Val when *Irf3*$^{-/-}$ BMDCs were infected with the 188-Val virus (Fig. 6). In support of our conclusion, Wu and colleagues recently reported ZIKV NS1 from Asian strain Z1106033 (isolated in 2015 from Suriname) inhibits IFN-β production through specific interaction with TBK1[34]; however, the authors did not compare the antagonistic activities of the NS1 proteins from different ZIKV strains and lineages.

On the crystal structure of ZIKV NS1, residue 188 is located at the β-roll domain of the dimeric NS1 interface[41]; it was previously shown that the A188V mutation does not change the NS1 dimerization[11]. Since NS1 is localized in the lumen of endoplasmic reticulum and secretory vesicles in a dimeric form[42], it remains to be determined where it interacts with TBK1 in the infected cellular compartments, and whether the NS1/TBK1 interaction is direct or through an intermediate host protein. Interestingly, the same A188V mutation of NS1 was recently shown to increase the extracellular secretion of NS1 that enhanced ZIKV infection of mosquitoes[11]. Taken together, the results support the mechanism that the current epidemic strains have accumulated the NS1 A188V mutation to directly enhance mosquito infection as well as to block mammalian immune response. Along a similar concept, mutations in the envelope genes (e.g., the A226V mutation in E1 protein) of chikungunya virus were previously found to enhance viral transmission by *A. albopictus* through increased infection of epithelial cells in the midgut, leading to re-emergence of large outbreaks[43,44].

In agreement with the role of NS1 A188V mutation in conferring antagonism of IFN-β induction, we observed increased viral replication and possibly neurovirulence after the FSS13025 strain was engineered with this mutation. When infecting immunocompetent BMDCs, FSS13025 A188V and PRVABC-59 WT viruses replicated to higher levels than their corresponding FSS13025 WT and PRVABC-59 V188A viruses, respectively (Fig. 6e); this result is further supported by the higher viremia and spleen viral loads when WT C57BL/6J mice were infected with FSS13025 A188V and PRVABC-59 WT viruses (Fig. 7e, f). In neonate CD-1 mice, intracranial infection with FSS13025 A188V resulted in a slightly higher mortality rate than the FSS13025 WT virus (Fig. 7c); the A188V mutant virus generated significantly higher brain viral loads than the WT

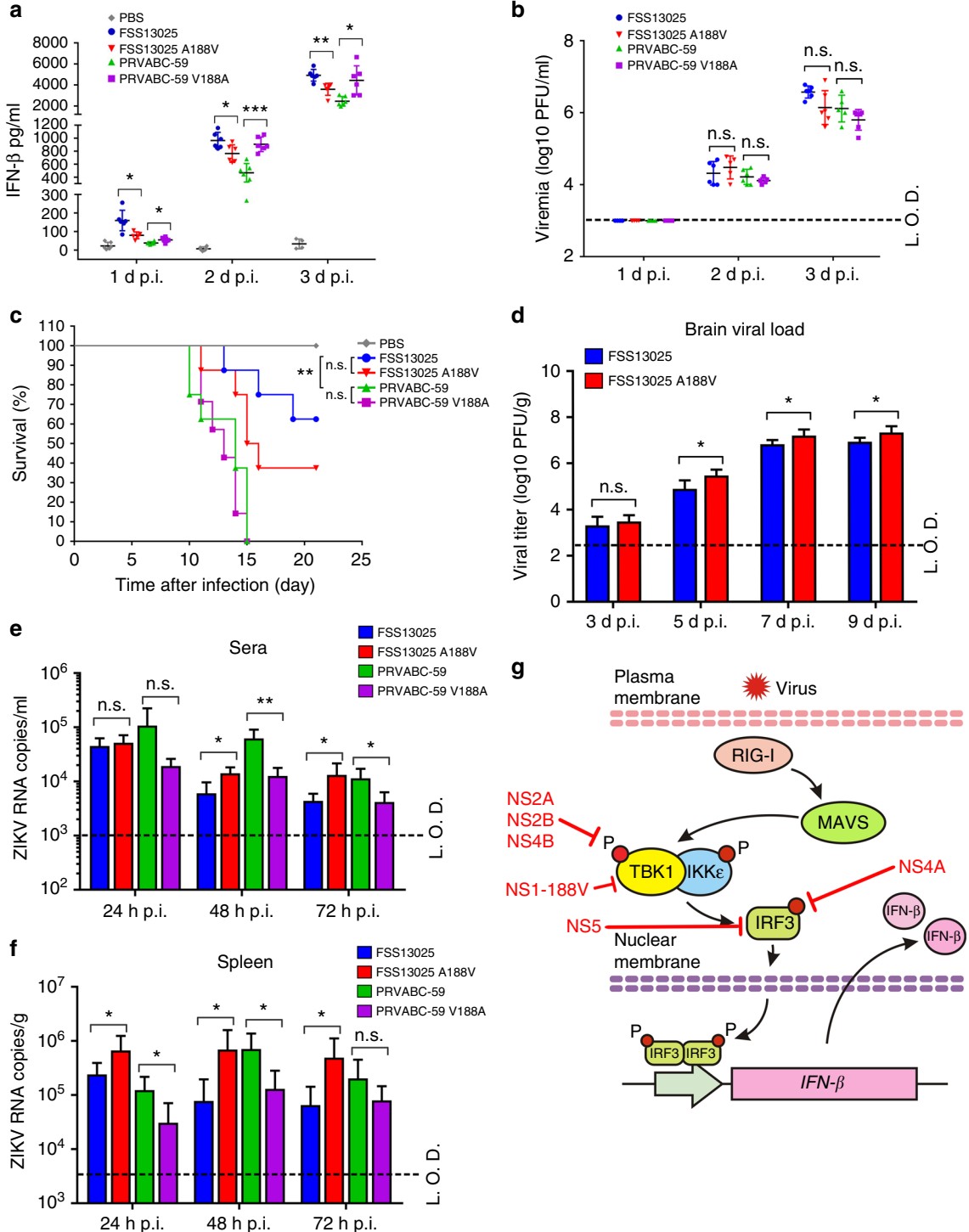

**Fig. 7** IFN-β protein expression in A129 mice and neurovirulence analysis in neonate CD-1 mice. Five groups of three-week-old A129 mice (*n* = 6) were intraperitoneally infected with 1×10⁵ PFU of FSS13025 WT, FSS13025 A188V, PRVABC-59 WT, PRVABC-59 V188A virus, or phosphate-buffered saline (PBS). Mouse blood were collected at days 1, 2, and 3 p.i., sera were separated and applied for IFN-β measurement by ELISA (**a**), and viremia were determined by plaque assay on Vero cells (**b**). **c** Comparison of neurovirulence of FSS13025 WT, FSS13025 A188V, PRVABC-59 WT, and PRVABC-59 V188A virus in CD-1 newborn mice. Groups of one-day-old CD-1 mice (*n* = 8 per group) were infected with 1×10³ PFU through the intracranial route. Survival curves are presented. **d** Two groups of one-day-old CD-1 mice (*n* = 5) were infected with 1×10³ PFU of FSS13025 WT and its corresponding A188V viruses through the intracranial route. Mouse brains were harvested at days 3, 5, 7, and 9 post-infection. Viral loads in brain were determined by plaque assay on Vero cells. **e**, **f** Four groups of C57BL/6J mice (*n* = 4 per group) were intraperitoneally infected with 1×10⁵ PFU of FSS13025 WT, FSS13025 A188V, PRVABC-59 WT, or PRVABC-59 V188A virus. Mouse blood and spleen were collected at 24, 48, and 72 h post-infection. Viral RNA in sera and spleens were extracted and quantified by qRT-PCR. Data are shown as means ± SD, *P < 0.05, **P < 0.01, or no significance (n.s.) through Student's two-sided *t*-test. L.O.D., limit of detection. **g** Summary of ZIKV non-structural proteins targeting different components of the RIG-I pathway. See text for details

FSS13025 virus (Fig. 7d). However, in the context of PRVABC-59, no difference in neurovirulence was observed in the WT and V188A viruses. Sequence alignment showed a total of 14 amino acid differences (all located outside the NS1 gene) between the FSS13025 and PRVABC-59 strains (Supplementary Table 1). These amino acid differences may mask the phenotype of the NS1 188 substitution in the context of PRVABC-59. Indeed, PRVABC-59 exhibited significantly higher neurovirulence than FSS13025 in the neonate mice (Fig. 7c). Experiments are ongoing to determine which of these amino acid variants are responsible for the neurovirulence difference between the pre-epidemic and epidemic strains. These mutations, together with NS1 A188V, may increase the epidemic potential and human diseases. Importantly, the result of lower neurovirulence of the pre-epidemic FSS13025 also supports the use of this strain for development of live-attenuated ZIKV vaccine[39,45].

It should be noted that African strains of ZIKV also have the NS1 188V residue[11]. Apparently when ZIKV reached Asia, an NS1 V188A substitution occurred, followed by reversion just before the recent outbreaks. If the NS1-188A residue has a selective disadvantage for transmission, this could have occurred through a founder effect when a single infected traveler introduced ZIKV to Asia. Similar evidence of dramatic results from founder effects have been reported for CHIKV[46]. Sequence alignment also reveals eight amino acid differences (including the A188V change) between the Asian FSS13025 and African Dakar-41525 strains (Supplementary Fig. 2). Functional analysis showed that NS1 from Dakar-41525 could also inhibit IFN-β production (Fig. 3a). In alignment with these results, recent in vivo and in vitro studies demonstrated that the African strains have higher fitness and virulence in vertebrates and mosquito vectors than the Asian strains[31,47–50]. Why are the higher fitness of African strains in mosquitoes and virulence in mice not correlated with higher infection incidents and disease severity in humans? Since lack of ZIKV diagnostics and surveillance in Africa limits our understanding of epidemiology and vector transmission there, more field and clinical studies are needed to understand this disconnection.

Besides accumulating mutations in the viral genome, two other non-exclusive mechanisms could also drive the recent emergence of ZIKV. (i) Stochastic introduction of ZIKV into a population lacking herd immunity, leading to increased incidence of infection with sufficient statistical power to detect rare clinical syndromes such as microcephaly and Guillain-Barré syndrome[51]. In support of this hypothesis, the recent increase of seroprevalence in the Americas is likely responsible for the decrease of ZIKV cases in 2017 (https://www.cdc.gov/zika/reporting/2017-case-counts.html). However, the threshold of seroprevalence needed to reduce the efficiency of ZIKV transmission in human populations remains unknown. (ii) Previous infection with DENV may exacerbate or protect against ZIKV disease because the two viruses share approximately 43% amino acid identity and extensive antibody cross-reactivity[52,53]. Antibodies against DENV and other flaviviruses were shown to enhance ZIKV infection in Fc receptor-bearing cells and in mice[54–56]. However, pre-existing DENV immunity did not result in more severe ZIKV disease in rhesus macaques[57]. Epidemiology studies are needed to determine whether pre-immune antibodies would cross-enhance different members from Flavivirus genus in humans. The answer to this question is critical to guide ZIKV vaccine and therapeutics antibody development.

In summary, we have identified five non-structural proteins of ZIKV that can antagonize IFN-β induction through targeting distinct components from the RIG-I pathway (Fig. 7e). For each of these proteins (except NS1), variants from the African, pre-epidemic Asian, and epidemic Asian strains were equally capable of inhibiting IFN-β production. In addition, we have uncovered an evolutionary mutation that confers NS1 to antagonize IFN-β

induction through modulating its binding to TBK1 (Fig. 7g). The functional effect of the NS1 mutation on viral replication and possibly neurovirulence in mice as well as mosquito infection supports its role in enhancing the recent epidemics and pathogenic potential of ZIKV.

## Methods

**Cells and viruses.** HEK-293T (CRL-11268), Vero (CCL-81), and JEG-3 (HTB-36) cells were purchased from the American Type Culture Collection (ATCC, Bethesda, MD) and maintained in high-glucose Dulbecco's modified Eagle's medium (DMEM), containing 10% fetal bovine serum (FBS; HyClone Laboratories, South Logan, UT) and 1% penicillin/streptomycin at 37°C in a 5% $CO_2$ incubator. A549 (CCL-185; ATCC) cells were grown in minimum essential medium, containing 1% nonessential amino acids, 1% sodium pyruvate, 10% FBS, and 1% penicillin/streptomycin at 37°C with 5% $CO_2$. ZIKV Cambodian strain (FSS13025, GenBank: KU955593.1), Puerto Rico strain (PRVABC-59, GenBank: KU501215), and the NS1 188 mutant viruses were prepared from a previously reported infectious cDNA clones[35]. The ZIKV Dakar-41525 (GenBank: KU955591.1) strain was obtained from the World Reference Center of Emerging Viruses and Arboviruses (WRCEVA) at the University of Texas Medical Branch.

**Plasmids and reagents.** All primers used for plasmid construction are listed in Supplementary Table 2. Expression plasmid pXJ was used to clone individual ZIKV genes from FSS13025, PRVABC-59, and Dakar-41525 strains. The pXJ plasmid was described previously[58]. Each gene was derived from RT-PCR using viral RNA or PCR from infectious cDNA clone plasmid[35,59], and was fused with a C-terminal HA-tag. The reporter plasmid pIFN-β-luc as well as the expression plasmids for RIG-I(2CARD), MAVS, IKKε, TBK1, and IRF3 were reported previously[60]. Internal control plasmid phRLuc-TK was purchased from Promega (Cat #E2241, Madison, WI). Anti-HA-tag (Cat #H6908, 1:1000), anti-Flag-tag (Cat #F7425, 1:1000), and anti-GAPDH (Cat #G9545, 1:1000) antibodies were purchased from Sigma-Aldrich (St. Louis, MO). Anti-pIRF3 at Ser-396 (Cat #29047, 1:1000), anti-IRF3 (Cat #10949, 1:1000), anti-TBK1 (Cat #3504, 1:1000), anti-pTBK1 at Ser-172 (Cat #5483, 1:1000) antibodies were purchased from Cell Signaling Technology (Danvers, MA). Anti-IKKε (Cat #ab7891, 1:1000) was from Abcam (Cambridge, MA), and anti-pIKKε at Ser-172 (Cat #06-1340, 1:1000) was purchased from Millipore (Darmstadt, Germany). Poly I:C (Cat #P9582) was from Sigma-Aldrich.

**Luciferase reporter assays.** HEK-293T cells were co-transfected in a 24-well plate ($1 \times 10^5$ cells per well) with 10 ng of IFN-β promoter reporter plasmid, 4 ng of renilla luciferase, and 20–80 ng viral protein expression plasmid using X-tremeGENE™ 9 (Roche, Mannheim, Germany) with a ratio 1:2. Empty pXJ vector was used to ensure the same total amount of plasmids in each well. For stimulation, 4 ng of plasmids RIG-I(2CARD), MAVS, IKKε, TBK1, or IRF3/5D was co-transfected, or cells were stimulated with 2 µg poly I:C each well for 16 h. At 24 h post-transfection, the transfected cells were lysed and dual-luciferase reporter assays were performed according to the manufacturer's instructions (Promega, Madison, WI).

**Co-immunoprecipitation and western blotting.** For immunoprecipitation, transfected HEK-293T cells were harvested in IP lysis buffer [20 mM Tris, 100 mM NaCl, 0.05% n-Dodecyl β-D-maltoside (Sigma-Aldrich), and protease inhibitor cocktail (Roche)], followed by immunoprecipitation using anti-HA magnetic beads (Pierce, Rockford, IL) or PureProteome magnetic beads (Millipore) according to the manufacturer's instructions. For western blotting, proteins were resolved by SDS-polyacrylamide gel electrophoresis and transferred onto a PVDF membrane using Trans-Blot Turbo Transfer System (BioRad, Hercules, CA), followed by a primary antibody as indicated and a secondary anti-Rabbit/Mouse IgG-Peroxidase antibody (Sigma-Aldrich). SuperSignal Femto Maximum Sensitivity Substrate (ThermoFisher, Rockford, IL) was used for protein visualization.

**Quantitative reverse transcription-PCR (qRT-PCR).** Total intracellular RNA was prepared using Trizol (Invitrogen, Carlsbad, CA) or RNeasy kit (Qiagen, Valencia, CA), and extracellular RNA was prepared using QIAamp viral RNA minikit (Qiagen) according to the manufacturer's instructions. qRT-PCR assays were performed using QuantiTect Probe RT-PCR kit (Qiagen) or iScript SYBR Green One-Step kit (BioRad), and a LightCycler 480 system (Roche, Basel, Switzerland) following the manufacturer's protocols. All primers and probes are listed in Supplementary Table 3.

**Preparation of NS1 recombinant ZIKVs.** Two infectious cDNA clones, one for FSS13025 and one for PRVABC-59 ZIKV[35,59], were used to perform site-directed mutagenesis for NS1 A188V and V188A changes, respectively. Mutations were introduced by overlap PCR, and the amplicon were inserted to the cDNA clones through utilizing the inner restriction enzyme sites of the vectors. Plasmids were linearized and viral RNAs were in vitro transcribed as described previously[35,59].

Equal amounts of ZIKV RNAs were electroporated into Vero cells, respectively. Virus were harvested at 96 h post-transfection.

**Plaque assay.** Viral titers were determined by plaque assay on Vero cells as previously described[61]. Briefly, viral samples were 10-fold serially diluted in DMEM with 2% FBS plus 1% penicillin/streptomycin and 100 μl of each dilution was added to Vero cells in a 24-well plate. The plates were incubated for 1 h and swirled every 15 min to ensure complete infection, after which the inoculum was removed. Five hundred microliters of methyl cellulose overlay containing 2% FBS was added to each well, and the plates were incubated at 37°C for 4 days. The plates were then fixed with 3.7% formaldehyde for 20 min and stained with 1% crystal violet for 1 min. Visible plaques were counted to determine viral titers as plaque forming units per ml (PFU/ml).

**Infection and IFN-β mRNA quantification in dendritic cells.** BMDCs were generated as described previously[62]. Briefly, bone marrow cells from WT C57BL/6J mice and Irf3$^{-/-}$ mice (bred in pathogen-free mouse facilities at UTMB) were isolated and cultured for 6 days in RPMI-1640 supplemented with granulocyte–macrophage colony stimulating factor and interleukin-4 (PeproTech, Rocky Hill, NJ) to generate myeloid dendritic cells. The Irf3$^{-/-}$ mice were reported previously[63]. The day 6-cultured BMDCs were infected with viruses at an MOI of 0.05. At day 1, 2, and 4 p.i., the infected cells were extracted for total intracellular RNA using Trizol (Invitrogen), and quantified for IFN-β mRNA and viral RNA levels by qRT-PCR. The extracellular RNA from culture fluids were isolated using QIAamp viral RNA Minikit (Qiagen), and quantified for viral RNA levels.

**Quantification of IFN-β protein production.** IFN-β protein levels in mouse sera were determined by ELISA (PBL Assay Science, Piscataway, NJ) following the manufacturer's instructions. Briefly, 3-week-old A129 mice were intraperitoneally infected with $1 × 10^5$ PFU viruses. Mouse sera were collected at 1, 2, and 3 days p.i., and quantified for the IFN-β protein level. Absorbance at 450 nm was measured using a Cytation 5 cell Imaging Multi-Mode Reader (BioTek, Winnooski, VT). IFN-β concentrations were calculated by fitting the standard curve of mouse IFN-β protein.

**A129, CD-1, and C57BL/6J mouse experiments.** Mouse studies were performed in accordance with the recommendations in the Guide for the Care and Use of Laboratory Animals of the National Institutes of Health. The protocols were approved by the Institutional Animal Care and Use Committee (IACUC) at the University of Texas Medical Branch (UTMB; Protocol Number 0209068B). The A129 mice (both genders; three-week-old) were bred in pathogen-free mouse facilities at UTMB, as previously described[37]. To measure brain viral loading, we infected one-day-old CD-1 mice ($n = 5$; Charles River Laboratories) with $10^3$ PFU of FSS13025 WT or FSS13025 NS1 A188V through the intracranial route. Mice brains were collect at days 3, 5, 7, and 9 p.i., and viral loads were measured by plaque assay as previous described[39]. For C57BL/6J mouse experiment, mice (female; six-week-old; $n = 4$ per group; the Jackson Laboratory) were infected with $10^5$ PFU of FSS13025 WT, FSS13025 NS1 A188V, PRVABC-59 WT, or PRVABC-59 NS1 V188A through the intraperitoneal route. Mouse blood and spleen were collected at days 1, 2, and 3 post-infection. Total RNAs were extracted using QIAamp viral RNA Minikit (Qiagen). Viral RNA was quantified by QuantiTect Probe RT-PCR kit (Qiagen) and a LightCycler 480 system (Roche, Basel, Switzerland) following the manufacturer's protocol as described above.

**Data analysis.** All data were analyzed with GraphPad Prism v7.02 software. Data are expressed as the mean ± standard deviation (SD). Comparisons of groups were performed using Student's two-sided $t$-test. A $P$-value of <0.05 indicates statistically significant.

**Data availability.** All relevant data are available from the authors upon request.

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

## Acknowledgements

We thank Slobodan Paessler at University of Texas Medical Branch (UTMB) for providing *Irf3*−/− mice. We also thank other colleagues at UTMB for helpful discussions and support during the course of this study. P.-Y.S. lab was supported by University of Texas Medical Branch (UTMB) startup award, University of Texas STARs Award, CDC grant for the Western Gulf Center of Excellence for Vector-Borne Diseases, Pan American Health Organization grant SCON2016-01353, the Kleberg Foundation Award, UTMB CTSA UL1TR-001439, and NIH grant AI127744. This research was also partially supported by NIH grants AI120942 and AI099123 to S.C.W and T.W., respectively. R.R. lab is supported by NIH grants K12HD052023 from ORWH and NICHD, and NIH/NIAID R21 AI132479-01. P.F.C.V. was supported by projects of CAPES (Zika Fast-Track) and CNPq grants 440405/2016-5 and 303999/2016-0 from the Ministry of Science and Technology of Brazil and by the Ministry of Health.

## Author contributions

H.X., H.L., C.S., A.E.M., B.T.D.N., D.B.A.M., J.Z., X.X. and M.I.G. conducted the experiments. H.X., C.S., X.X., P.F.C.V., S.C.W., T.W., R.R. and P.-Y.S. designed the experiments, analyzed the data, and wrote the paper.

## Additional information

**Competing interests:** The authors declare no competing financial interests

