## [Peer Review File · Nature Communications]

Reviewers' comments:

Reviewer #1 (Remarks to the Author):

Xia et al examine the inhibitory effects of Zika virus (ZIKV) nonstructural (NS) proteins on antiviral signaling pathways initiated through RIG-I-like helicase (RLR) ligation. They find roles for multiple NS proteins in suppressing the IFN β signaling cascade, and find that NS2A, NS2B, and NS4B suppress IFN production by preventing TBK1 phosphorylation, NS4A appears to target IRF3 phosphorylation, and NS5 imparts a weak block to IRF3 signaling through direct binding without suppressing IRF3 phosphorylation. Interestingly, the NS1 derived from a pre-epidemic Asian lineage virus (Cambodia 2010) had no measurable effects on these signaling pathways, whereas NS1 from 1984 African strain or an epidemic Asian lineage virus (Puerto Rico 2015) demonstrated IFN antagonism associated with a differential ability to bind TBK1. The mutation responsible for acquired TBK1 binding is the same one previously shown to confer increased NS1 accumulation in sera of infected mice that enhances mosquito acquisition of ZIKV. Mutation of the residue at NS1 position 188 in recombinant viruses did not directly affect virus replication, but did modulate circulating IFN β in the sera of infected AG129 mice, suggesting that this mutation may be biologically significant in virus virulence independent of mosquito acquisition. However, the biological consequences of the differences in IFN are not clearly elucidated.

Major comments:

1. NS1 is a secreted protein and Liu et al 2017 has already demonstrated that residue 188 is important for NS1 accumulation in the sera of infected mice. Although somewhat controversial, flavivirus NS1 has been shown to either activate toll-like receptor (TLR) signaling or suppress it (e.g. Modhiran et al 2017 Immunol Cell Biol; Chen et al. 2015 PLoS Path; Crook et al 2014 Virology). Therefore, the effects of this NS1 mutation on the ability of secreted NS1 to modulate TLR signaling should be examined, potentially by infecting TRIF/MyD88 double knock out mice which should retain differences in circulating IFN. Without this data, it is difficult to conclude that the role of NS1 in vivo in IFN β expression is exclusively through intracellular interactions with TBK1.
2. Infection of WT mice should be performed with the various viruses. Although the WT mouse has known limitations, it is difficult to understand what the biological consequences of these relatively small differences in circulating IFN might be. A clear peak of virus replication can be measured in the spleen in the first 2-3 days of infection in WT mice, and this should be sufficient to measure differences in virus replication in a model where IFN signaling is intact if the differences in circulating IFN are meaningful.

Other comments:

1. Why wasn't NS2B/3 included in this analysis? This is a major potential antagonist of IFN signaling, and may reveal differences between viruses that are not observed when NS2B or NS3 are expressed alone.
2. Figure 2D: Interactions between NS5 and IRF3. No controls are included for specific pull down. NS5 is notoriously sticky and is easy to non-specifically precipitate from lysates. An Ig control needs to be included to ensure the interaction is not a false positive.
3. The interactions between the various NS1 proteins and 188 mutants should be examined

by IFA colocalization. This would provide support for the major finding that some NS1s interact with TBK while others do not, which is only demonstrated through 1 experiment.

4. Neurovirulence in neonatal mice: Why was the dose of 10^3 chosen? This seems very high and would likely negate any effects that could be observed with a lower dose. Where 100 and 10 pfu trialed as well?

Minor comments:

Line 123: This conclusion should be 'at or downstream of RIG-I'

Reviewer #2 (Remarks to the Author):

Shi et al. report that several ZIKV non-structural proteins limit the host interferon response in infected cells. The authors further report that among these non-structural proteins, NS1 stands out because of an acquired mutation (A188V) that is present in epidemic strains. The authors propose that this acquired mutation allows the virus to evade immune responses, potentiating infections and epidemics.

The experimental approach to the work is thorough. A number of biochemical and in vivo approaches are used to generate data presented in the manuscript.

Figure 1: These results are consistent with findings by Wu et al. (ref. 34), although Wu et al did not compare pre-and post-epidemic viruses. The figure legend should state that the viral proteins were cloned from the pre-epidemic, Cambodian strain ZIKV (FSS13025) isolated from a patient in 2010. The title of the manuscript seems to focus on the NS1 findings (without naming), yet Figures 1 and 2 emphasize other non-structural proteins. Because the findings in Figs 1 and 2 overlap with Wu et al (ref 34), the presentation could be sharpened by applying consistency between the title and the dataset presented. Pointing to the novelty of the NS1 differences between pre and post epidemic strain at an earlier point in the presentation (it is now in Fig 3) would focus on novel findings. As now presented it may seem odd to the reader that, if the paper emphasizes NS1, that in Fig. 1: i) there is no change by NS1 expression in panel A (b/c it is the FSS13025 strain), and ii) the effects of NS1 expression are not presented in panels c-f. The summary here is that the presentation seems to show some ambivalence about defining the main point of paper (NS1 and evolutionary mutations or effects of ZIKV proteins on interferon?).

Fig 2: Labeling the expressed proteins at the top of the figure as HA-NS2A, HA-NS2B would help the reader interpret the data. The data here seem solid, but the purpose of including MTase and RdRp and NS5 effects on RIG-I induced IFN activity in an NS1-focused paper is not clear as presented. Are these results presented to show that other non-structural proteins limit host response, or that the other (non-NS1) non-structural proteins behave similarly across viruses?

Fig. 3: In panel (a), the meaning of the Y-axis label ('% fold induction) is not clear. The

method of normalization and definition of “% fold induction” should be clarified. The text states that Remarkably, NS1 from both PRVABC-59 and Dakar-41525 stains 160 suppressed IFN- β induction by 60% and 75% . My interpretation of the results is that the PRVABC-59 expression resulted in about a 40% decrease, and the Dakar resulted in a 25-30% decrease of “fold induction”. The correlation between the text and the figure should be clarified. The data support the conclusion that NS1 activity in reducing IFN- β promoter activity shows inter-virus strain differences that are not observed with the other non-structural proteins. However, the data in Figure 3a seem to show that the observed effect of NS1 is less than described in the text.

Fig. 4: The data in Figure 4 are the most compelling results presented in the manuscript. These data very strongly suggest that NS1 188V binds to TBK1 (through IP pulldown experiments) and that the interaction correlates with dramatically diminished IRF3 and TBK1 phosphorylation. The IPs and WB in panels e and f are strikingly clean and suggest not just a reduction in phosphorylation, but the absence of phosphorylation. In light of the very strong impact of NS1 188V on blocking TBK phosphorylation, an expected parallel result would have been to observe a much more dramatic effect on IFNB in Figure 3.

Fig. 5: The authors use mutant viruses and present data suggesting that the effects of NS1 188V on innate immune signaling are observed in two different immune competent cells , i.e. A549 and JEG-3. What are the units on the Y-axis of panel 5d (“relative viral RNA”)? The results in 5c suggest that the NS1 188V position correlates with \sim 5-fold differences in IFNB mRNA levels at 24 hours with the PRVABC virus without a corresponding difference in the level of viral RNA.

Fig 6: The authors quantify IFNB mRNA levels in BMDD cells and show that there are modest, but statistically significant decreases in IFNB mRNA levels, accompanying infection by the 188V virus, with corresponding increases in converting 188V to 188A.

Fig. 7. The authors extend the biochemical and cell culture experiments to examine IFN- β protein expression in A129 mice and neurovirulence analysis in neonate CD-1 mice. The results suggest that there are modest changes in IFNB mRNA levels that parallel the results observed using other assays. The neurovirulence experiment (panel c) did not show statistical significance in attempts to demonstrate that converting the NS1 188A virus to 188V decreased survival. The box legend in panel 7c shows “**” statistical significance in comparing PRVAB-59 with FSS13025. However, the table in panel d indicates that the average survival time in comparing these two viruses is “n.s.”. Why is there a discrepancy between the two?

Summary. This manuscript addresses important problems in infectious diseases and host responses that are of broad interest to readership. The authors apply multiple experimental techniques (biochemical, molecular, whole animal) to test their hypotheses. The data are of very high quality. What is novel here is that different flaviviruses have mutations that correlate with evasion of host responses. The authors refer to these changes as evolutionary, with connections between NS1 mutations, improved evasion of host responses, and the appearance of epidemic strains.

Though of very high quality, the impact of the paper in its current form is somewhat limited in several ways that may be addressable. The comparative NS1 data should appear earlier in the paper if the focus is indeed on the NS1 mutation. Second, the differences in IFNB mRNA levels and protein levels, as a function of the NS1 188V position, seem modest (Fig. 3, 5, 6). It is admittedly difficult to correlate specific mRNA levels with a physiological innate immune response. Nonetheless, the dramatic effects on IRF3 and TBK1 phosphorylation (Fig. 4) do not seem to be reflected directly in the IFNB promoter response data (Fig 1,3). In addition, the neurovirulence with the 188V mutations results did not reach statistical significance (Fig. 7). With such a dramatic effect on TBK1 and IRF3 phosphorylation, it is puzzling why the impact on IFNB mRNA and IFNB promoter responses is not more dramatic. Addressing this differential seems critical to an argument stating that a single mutation in NS1 is at the heart of host immune evasion.

The author's idea to compare pre-epidemic and post-epidemic viruses is clever; however, the NS1 story is complicated by the finding that the Dakar pre-epidemic strain already has the NS1 188V mutation. The authors speculate that upon reaching Asia, the virus mutated to 188A, and then reverted to 188V at the time of the 2015 outbreak. This is plausible, but seems difficult to evaluate. Finally, the results in this paper and those of Wu (ref 34) suggest that the combined effects of amino acid substitutions in multiple non-structural proteins collectively form the basis of host immune evasion. In the end, all roads do not seem to point definitively to only NS1 188V, as implied in the title.

Minor points

Line 159: "stains" should be strains

Line 278: "secretary" should be secretary

Line 597: "inhibits" should be inhibit

Reviewer #1

Xia et al examine the inhibitory effects of Zika virus (ZIKV) nonstructural (NS) proteins on antiviral signaling pathways initiated through RIG-I-like helicase (RLR) ligation. They find roles for multiple NS proteins in suppressing the IFN β signaling cascade, and find that NS2A, NS2B, and NS4B suppress IFN production by preventing TBK1 phosphorylation, NS4A appears to target IRF3 phosphorylation, and NS5 imparts a weak block to IRF3 signaling through direct binding without suppressing IRF3 phosphorylation. Interestingly, the NS1 derived from a pre-epidemic Asian lineage virus (Cambodia 2010) had no measurable effects on these signaling pathways, whereas NS1 from 1984 African strain or an epidemic Asian lineage virus (Puerto Rico 2015) demonstrated IFN antagonism associated with a differential ability to bind TBK1. The mutation responsible for acquired TBK1 binding is the same one previously shown to confer increased NS1 accumulation in sera of infected mice that enhances mosquito acquisition of ZIKV. Mutation of the residue at NS1 position 188 in recombinant viruses did not directly affect virus replication, but did modulate circulating IFN β in the sera of infected AG129 mice, suggesting that this mutation may be biologically significant in virus virulence independent of mosquito acquisition. However, the biological consequences of the differences in IFN are not clearly elucidated.

Response: We have performed two sets of new experiments to bolster the biological relevance of the NS1 A188V mutation. (i) We infected one-day-old CD-1 mice with FSS13025 WT and its NS1 A188V viruses. The NS1 A188V mutant virus showed significantly higher viral loads in the brains of infected mice than the WT FSS13025 virus did. The new results are presented in Fig. 7d. (ii) We infected wild-type C57BL/6J mice which are immunocompetent for interferon production and signaling. For both FSS13025 and PRVABC59 strains, viruses with NS1 188-Val exhibited higher levels of viral RNA in both sera and spleen than their corresponding NS1 188-Ala viruses. These new results are now presented in Figs. 7e & f.

Major comments:

1. NS1 is a secreted protein and Liu et al 2017 has already demonstrated that residue 188 is important for NS1 accumulation in the sera of infected mice. Although somewhat controversial, flavivirus NS1 has been shown to either activate toll-like receptor (TLR) signaling or suppress it (e.g. Modhiran et al 2017 Immunol Cell Biol; Chen et al. 2015 PLoS Path; Crook et al 2014 Virology). Therefore, the effects of this NS1 mutation on the ability of secreted NS1 to modulate TLR signaling should be examined, potentially by infecting TRIF/MyD88 double knock out mice which should retain differences in circulating IFN. Without this data, it is difficult to conclude that the role of NS1 in vivo in IFN β expression is exclusively through intracellular interactions with TBK1.

Response: We thank the reviewer for this interesting suggestion. We respectfully disagree with the reviewer for the following three reasons. (i) The NS1's role in activating or suppressing TLR signaling is highly controversial, as acknowledged by the reviewer. (ii) Controversial results were reported in dengue and West Nile viruses. So far, there has been no reports on the role of Zika NS1 in modulating TLR signaling. (iii) The TRIF/MyD88 double knockout mice are not available in my and collaborators' labs. Even if we get these mice, it will take two months to expand the mouse colony before we could start the experiments. Instead, we have performed

two sets of new experiments to bolster the biological relevance of the NS1 A188V mutation, as described in the preceding section.

2. Infection of WT mice should be performed with the various viruses. Although the WT mouse has known limitations, it is difficult to understand what the biological consequences of these relatively small differences in circulating IFN might be. A clear peak of virus replication can be measured in the spleen in the first 2-3 days of infection in WT mice, and this should be sufficient to measure differences in virus replication in a model where IFN signaling is intact if the differences in circulating IFN are meaningful.

Response: Done. We have performed the suggested experiments in WT C57BL/6J mice. Six-week-old C57BL/6J mice were infected with 10^5 PFU of WT FSS13025, WT PRVABC-59, and their corresponding NS1 mutant viruses through the intraperitoneal route. Blood and spleen were collected at days 1, 2, and 3 post-infection. Viral RNA in sera and spleen were quantified by qRT-PCR. Viral RNA levels of FSS13025 A188V and PRVABC-59 WT were significantly higher than their corresponding WT and V188A viruses in both sera and spleen. These new results are presented in Fig. 7e & f.

Other comments:

1. Why wasn't NS2B/3 included in this analysis? This is a major potential antagonist of IFN signaling, and may reveal differences between viruses that are not observed when NS2B or NS3 are expressed alone.

Response: Done. We have now added the results with NS2B/3 construct. Upon poly I:C treatment, the expression of NS2B/3 proteins did not significantly suppress IFN- β induction. The new result has been added to Supplementary Fig. 1b.

2. Figure 2D: Interactions between NS5 and IRF3. No controls are included for specific pull down. NS5 is notoriously sticky and is easy to non-specifically precipitate from lysates. An Ig control needs to be included to ensure the interaction is not a false positive.

Response: Done. We have added the suggested negative control in Fig. 2d (last lane).

3. The interactions between the various NS1 proteins and 188 mutants should be examined by IFA colocalization. This would provide support for the major finding that some NS1s interact with TBK while others do not, which is only demonstrated through 1 experiment.

Response: We have performed the confocal imaging experiment. Unfortunately, we observed no or very low level of co-localization of NS1 and TBK1 (see the results below). We currently don't know what caused the discrepancy between the confocal imaging result and the co-immunoprecipitation result. However, in agreement with our co-immunoprecipitation result, Wu and co-workers recently showed that ZIKV NS1 with 188-Val interacted with TBK1, leading to inhibition of TBK1 dimerization and phosphorylation (*Cell Discov* **3**, 17006, doi:10.1038/celldisc). We have mentioned these results in Discussion.

4. Neurovirulence in neonatal mice: Why was the dose of 10^3 chosen? This seems very high and would likely negate any effects that could be observed with a lower dose. Where 100 and 10 pfu trialed as well?

Response: The selection of 10^3 PFU dose in the neurovirulence in neonatal mice was based on our previous work (Nat. Med. 2017 doi:10.1038/n 533 m.4322 and Nat. Commun. 2017 doi: 10.1038/s41467-017-00737-8). Intracranial injection of 10^3 PFU of FSS13025 strain in the one-day-old mice lead to about 37.5% mortality; therefore, this dose is appropriate to analyze the effect of 188-Val on neurovirulence; indeed, the A188V mutation increased the mortality rate to 62.5% (Fig. 7c). In addition, we have now compared the viral replication levels of WT FSS13025 and its corresponding A188V mutant viruses in the brains of infected animals. The results showed that the A188V mutant replicated to significantly higher levels in the brain than the WT FSS13025 virus at days 5-9 post-infection. The new results are presented in Fig. 7d.

Minor comments:

Line 123: This conclusion should be 'at or downstream of RIG-I'

Response: Corrected.

Reviewer #2

Shi et al. report that several ZIKV non-structural proteins limit the host interferon response in infected cells. The authors further report that among these non-structural proteins, NS1 stands out because of an acquired mutation (A188V) that is present in epidemic strains. The authors propose that this acquired mutation allows the virus to evade immune responses, potentiating infections and epidemics. The experimental approach to the work is thorough. A number of biochemical and in vivo approaches are used to generate data presented in the manuscript.

Response: We thank the review for the positive comment.

Figure 1: These results are consistent with findings by Wu et al. (ref. 34), although Wu et al did not compare pre-and post-epidemic viruses. The figure legend should state that the viral proteins were cloned from the pre-epidemic, Cambodian strain ZIKV (FSS13025) isolated from a patient in 2010.

Response: Corrected. We have added “the viral proteins were cloned from the pre-epidemic, Cambodian strain ZIKV (FSS13025) isolated from a patient in 2010” to the figure legend.

The title of the manuscript seems to focus on the NS1 findings (without naming), yet Figures 1 and 2 emphasize other non-structural proteins. Because the findings in Figs 1 and 2 overlap with Wu et al (ref 34), the presentation could be sharpened by applying consistency between the title and the dataset presented. Pointing to the novelty of the NS1 differences between pre and post epidemic strain at an earlier point in the presentation (it is now in Fig 3) would focus on novel findings. As now presented it may seem odd to the reader that, if the paper emphasizes NS1, that in Fig. 1: i) there is no change by NS1 expression in panel A (b/c it is the FSS13025 strain), and ii) the effects of NS1 expression are not presented in panels c-f. The summary here is that the presentation seems to show some ambivalence about defining the main point of paper (NS1 and evolutionary mutations or effects of ZIKV proteins on interferon?).

Response: We thank the reviewer for this thoughtful suggestion. The goals of this study are to identify (i) ZIKV non-structural proteins that can antagonize host type-I IFN production and (ii) mutations in the recent epidemic strains that modulate ZIKV evasion of type-I IFN induction. To clarify these two goals, we have now added a section on “Overall experimental approach” at the beginning of Results. Therefore, we keep the original order of the figures.

Fig 2: Labeling the expressed proteins at the top of the figure as HA-NS2A, HA-NS2B would help the reader interpret the data. The data here seem solid, but the purpose of including MTase and RdRp and NS5 effects on RIG-I induced IFN activity in an NS1-focused paper is not clear as presented. Are these results presented to show that other non-structural proteins limit host response, or that the other (non-NS1) non-structural proteins behave similarly across viruses?

Response: We have revised the labels as suggested by the reviewer. As indicated in the response to the preceding comment, we have two goals to achieve in this study, one of which is to identify ZIKV non-structural proteins that can antagonize host type-I IFN production. It is therefore important to dissect the MTase and RdRp domains for their abilities to interact with IRF3.

Fig. 3: In panel (a), the meaning of the Y-axis label (“% fold induction”) is not clear. The method of normalization and definition of “% fold induction” should be clarified. The text states that Remarkably, NS1 from both PRVABC-59 and Dakar-41525 stains 160 suppressed IFN- β induction by 60% and 75%. My interpretation of the results is that the PRVABC-59 expression resulted in about a 40% decrease, and the Dakar resulted in a 25-30% decrease of “fold induction”. The correlation between the text and the figure should be clarified. The data support the conclusion that NS1 activity in reducing IFN-b promoter activity shows inter-virus strain differences that are not observed with the other non-structural proteins. However, the data in Figure 3a seem to show that the observed effect of NS1 is less than described in the text.

Response: Done. (i) In the figure legend, we have now added the information about how to calculate the fold induction for the Y-axis label. This is virtually the same as described in the legend to Fig. 1. (ii) We corrected the suppression levels to 40% and 30%, as pointed out by the reviewer.

Fig. 4: The data in Figure 4 are the most compelling results presented in the manuscript. These data very strongly suggest that NS1 188V binds to TBK1 (through IP pulldown experiments) and that the interaction correlates with dramatically diminished IRF3 and TBK1 phosphorylation. The IPs and WB in panels e and f are strikingly clean and suggest not just a reduction in phosphorylation, but the absence of phosphorylation. In light of the very strong impact of NS1 188V on blocking TBK phosphorylation, an expected parallel result would have been to observe a much more dramatic effect on IFNB in Figure 3.

Response: TBK1 is a key kinase in the IRF3-mediated IFN- β induction pathway, which can be recruited and activated by MAVS on the mitochondrial membrane and alternatively by STING on the ER membrane. TBK1 forms oligomerization and phosphorylate themselves to activate, and then phosphorylate IRF3 to stimulate the pathway. ZIKV NS1 (with 188-Val) interacts with TBK1 and block TBK1 oligomerization, which reduces TBK1 phosphorylation and subverts IFN- β induction. Since TBK1 can be activated by adaptor protein other than MAVS (such as STING on the ER membrane), ZIKV NS1 may not be able to suppress all those activation pathways; under those circumstances, TBK1 could still be partially activated in the presence of NS1 (with 188-Val).

Fig. 5: The authors use mutant viruses and present data suggesting that the effects of NS1 188V on innate immune signaling are observed in two different immune competent cells, i.e. A549 and JEG-3. What are the units on the Y-axis of panel 5d (“relative viral RNA”)? The results in 5c suggest that the NS1 188V position correlates with ~ 5-fold differences in IFNB mRNA levels at 24 hours with the PRVABC virus without a corresponding difference in the level of viral RNA.

Response: Corrected. We have run a standard curve and calculated the viral RNA copy numbers for Fig. 5d & f. We have now changed the labels of Y-axis to “Intracellular ZIKV RNA copies” in Fig. 5d & f.

Fig 6: The authors quantify IFNB mRNA levels in BMDC cells and show that there are modest, but statistically significant decreases in IFNB mRNA levels, accompanying infection by the 188V virus, with corresponding increases in converting 188V to 188A.

Response: Yes.

Fig. 7. The authors extend the biochemical and cell culture experiments to examine IFN- β protein expression in A129 mice and neurovirulence analysis in neonate CD-1 mice. The results suggest that there are modest changes in IFNB mRNA levels that parallel the results observed using other assays. The neurovirulence experiment (panel c) did not show statistical significance in attempts to demonstrate that converting the NS1 188A virus to 188V decreased survival. The box legend in panel 7c shows “***” statistical significance in comparing PRVAB-59 with FSS13025. However, the table in panel d indicates that the average survival time in comparing these two viruses is “n.s.”. Why is there a discrepancy between the two?

Response: We have deleted the non-significant data of average survival time. Instead, we have performed a new experiment to measure the viral loads in the brains after the one-day-old CD-1 mice were intracranially infected with WT FSS13025 and its corresponding A188V mutant viruses. The results showed that the A188V mutant virus replicated to higher levels in the brain than the WT FSS13025 did. The new results are presented in Fig. 7d.

Summary. This manuscript addresses important problems in infectious diseases and host responses that are of broad interest to readership. The authors apply multiple experimental techniques (biochemical, molecular, whole animal) to test their hypotheses. The data are of very high quality. What is novel here is that different flaviviruses have mutations that correlate with evasion of host responses. The authors refer to these changes as evolutionary, with connections between NS1 mutations, improved evasion of host responses, and the appearance of epidemic strains.

Response: We thank the review for the positive comment.

Though of very high quality, the impact of the paper in its current form is somewhat limited in several ways that may be addressable. The comparative NS1 data should appear earlier in the paper if the focus is indeed on the NS1 mutation.

Response: As indicated in previous section, we have added a section “Overall experimental approach” to the beginning of Results to clarify this point. We have two aims for this study: (i) Identify ZIKV non-structural proteins that can antagonize host type-I IFN production; and (ii) identify mutations in the recent epidemic strains that modulate ZIKV evasion of type-I IFN induction.

Second, the differences in IFNB mRNA levels and protein levels, as a function of the NS1 188V position, seem modest (Fig. 3, 5, 6). It is admittedly difficult to correlate specific mRNA levels with a physiological innate immune response. Nonetheless, the dramatic effects on IRF3 and TBK1 phosphorylation (Fig. 4) do not seem to be reflected directly in the IFNB promoter response data (Fig 1,3). In addition, the neurovirulence with the 188V mutations results did not reach statistical significance (Fig. 7). With such a dramatic effect on TBK1 and IRF3 phosphorylation, it is puzzling why the impact on IFNB mRNA and IFNB promoter responses is not more dramatic. Addressing this differential seems critical to an argument stating that a single mutation in NS1 is at the heart of host immune evasion.

The author’s idea to compare pre-epidemic and post-epidemic viruses is clever; however, the

NS1 story is complicated by the finding that the Dakar pre-epidemic strain already has the NS1 188V mutation. The authors speculate that upon reaching Asia, the virus mutated to 188A, and then reverted to 188V at the time of the 2015 outbreak. This is plausible, but seems difficult to evaluate. Finally, the results in this paper and those of Wu (ref 34) suggest that the combined effects of amino acid substitutions in multiple non-structural proteins collectively form the basis of host immune evasion. In the end, all roads do not seem to point definitively to only NS1 188V, as implied in the title.

Response: We thank the reviewer for these thoughtful comments. As indicated in the response to Reviewer #1, we have performed two sets of new experiments to bolster the biological relevance of the NS1 A188V mutation. (i) We infected one-day-old CD-1 mice with FSS13025 WT and its NS1 A188V viruses. The NS1 A188V mutant showed higher viral load in the brain than the WT FSS13025 virus. (ii) We infected wild-type C57BL/6J mice which are immunocompetent for interferon production and signaling. For both FSS13025 and PRVABC59 strains, viruses with NS1 188-Val exhibited higher levels of viral RNA in sera and spleen than their corresponding NS1 188-Ala viruses.

Minor points

Line 159: “stains” should be strains

Response: Corrected.

Line 278: “secretary” should be secretory

Response: Corrected.

Line 597: “inhibits” should be inhibit

Response: Corrected.

REVIEWERS' COMMENTS:

Reviewer #1 (Remarks to the Author):

The authors have addressed the comments of this reviewer to a satisfactory level.

Reviewer #2 (Remarks to the Author):

The authors have addressed all of the comments raised in my initial review. Additional experiments have been performed to support the major conclusions of the paper, and additional text has been added where the reviews suggested clarification. The response is quite satisfactory in responding positively to all points in the review.